# Using a Tip Characterizer to Investigate Microprobe Silicon Tip Geometry Variation in Roughness Measurements

**DOI:** 10.3390/s22031298

**Published:** 2022-02-08

**Authors:** Min Xu, Ziqi Zhou, Thomas Ahbe, Erwin Peiner, Uwe Brand

**Affiliations:** 1Physikalisch-Technische Bundesanstalt (PTB), Bundesallee 100, 38116 Braunschweig, Germany; thomas.ahbe@ptb.de (T.A.); uwe.brand@ptb.de (U.B.); 2Institute of Production Measurement Technology (IPROM), Technische Universität Braunschweig, Schleinitzstraße 20, 38106 Braunschweig, Germany; ziqi.zhou@tu-braunschweig.de; 3Institute of Semiconductor Technology (IHT), Technische Universität Braunschweig, Hans-Sommer-Straße 66, 38106 Braunschweig, Germany; e.peiner@tu-bs.de; 4Laboratory for Emerging Nanometrology (LENA), Langer Kamp 6 a/b, 38106 Braunschweig, Germany

**Keywords:** roughness measurement, piezoresistive microprobe, silicon fracture, tip characterization

## Abstract

Given their superior dynamics, microprobes represent promising probe candidates for high-speed roughness measurement applications. Their disadvantage, however, lies in the fact that the volume of the microprobe’s silicon tip decreases dramatically during roughness measurement, and the unstable tip geometry leads to an increase in measurement uncertainty. To investigate the factors that influence tip geometry variation during roughness measurement, a rectangular-shaped tip characterizer was employed to characterize the tip geometry, and a method for reconstructing the tip geometry from the measured profile was introduced. Experiments were conducted to explore the ways in which the tip geometry is influenced by tip wear, probing force, and the relative movement of the tip with respect to the sample. The results indicate that tip fracture and not tip wear is the main reason for tip volume loss, and that the lateral dynamic load on the tip during scanning mode is responsible for more tip fracture than are other factors.

## 1. Introduction

With the advantages of miniaturized size and high sensitivity, the silicon-based cantilever has become an important sensor [1,2]. It helps researchers to overcome technological obstacles encountered when using conventional technologies. Combined with diverse techniques of lithography, coating, and microengineering, various cantilevers have been developed and applied in diverse fields, such as topography mapping, material property characterization, or cell and virus particle detection.

High-speed surface roughness measurement is another potential application for the silicon-based cantilever. Surface roughness measurements allow us to predict the performance of a mechanical component, and they play an important role in component quality determination [3,4]. Such measurements demand a measurement technology that combines high accuracy with high throughput. Surface roughness can be measured by a contact stylus instrument or by an optical instrument, such as a vertical scanning interferometer (VSI). The lateral resolution of an optical instrument is limited by the objective’s numerical aperture and optical wavelength [5], and the signal quality is degraded by undesired light reflection and diffraction effects. It has been proven that the stylus method is more accurate than the VSI for hard material roughness measurements [6,7,8].

One drawback of the contact stylus instrument is its low throughput. Although the maximum traverse speeds of state-of-the-art stylus instruments are in the range of 1–3 mm/s [9], in many roughness measurements the stylus begins losing surface trackability at traverse speeds as low as 500 µm/s [10]. To ensure tracking fidelity at high traverse speeds and to increase the throughput of the contact stylus instrument, a probe with better dynamics is required.

The microprobes developed by the Physikalisch-Technische Bundesanstalt (PTB) together with the Institute for Semiconductor Technology of the Technical University of Braunschweig and the Forschungsinstitut für Mikrosensorik GmbH (CiS) Erfurt are slender piezoresistive monocrystalline silicon cantilevers [11,12]. They are commercially available as type CAN50-2-5 from CiS GmbH. The cantilever is 5 mm long, 200 µm wide, 50 µm thick and has a mass of about 0.1 mg. At the end of the cantilever, there is an integrated octagonal pyramidal silicon tip with a height of about 100 µm and a half opening angle of 20° (see Figure 1). The full bridge piezoresistive strain gauge on the back of the cantilever close to its clamping converts cantilever bending into a voltage output with a nonlinearity below 0.3% [13]. The microprobe demonstrates superior dynamics because of its low mass. Theoretical analysis and experimental results indicate that it can track surfaces with steep features up to traverse speeds of 10 mm/s with high fidelity [14]. When the probing force is larger than 28 µN, the microprobe can measure a surface of 10 µm amplitude and 11 µm wavelength with the traverse speed up to 15 mm/s without tip flight [15].

However, this microprobe runs into a problem that is considered the Achilles’ heel of cantilevers with integrated silicon tips: the stability of the tip’s geometry [15]. As shown in Figure 2, the radius of a new CAN50-2-5 microprobe tip is about 0.1 µm. After 300 m of roughness measurement, the tip end becomes flat and wide. Since the measured profiles and images are derived from the dilation of the tip and the artifact surface, any variation in tip geometry brings uncertainties to the measurement result (see Figure 3) [16,17].

The tip geometry can change as a result of tip wear, tip fracture, or contamination. The first two factors cause tip volume loss, while the last factor causes the tip volume to increase. In roughness measurements, the roughness standards are made primarily of hard materials and are for the most part kept clean before and during measurements. Our work, therefore, aims to find out the main reason behind tip volume loss in roughness measurements. This investigation will help users of microprobes to improve their measurement settings to protect the tip and will also seek to identify the direction that the development of a durable microprobe tip might take.

A tip characterizer with rectangular structures is employed in the experiments to characterize the tip geometry. In the following, the method of characterizing the tip geometry with a rectangular-shaped tip characterizer is explained, and experiments are performed to explore the factors that influence tip geometry variation during roughness measurements.

## 2. Tip Characterization

Three techniques have been proposed by researchers to characterize tip geometry:Direct tip imaging using electron microscopes, such as scanning electron microscopes (SEMs), transmission electron microscopes (TEMs), or scanning TEMs (STEMs) [18,19];Characterizing the tip with a tip characterizer [20,21,22];Blind reconstruction techniques without a calibrated tip characterizer [23]

The electron microscope can image the tip directly. However, the imaging can only be performed offline and with the cantilever demounted from the measurement instrument. Because our experiment demands that imaging be performed numerous times during roughness measurements, it is not realistic to rely solely on electron microscopes to undertake this task.

Tip characterizers are artifacts with surface features specially designed for tip characterization. Characterizing the tip with a tip characterizer is assumed to achieve higher precision than the blind reconstruction method and is expected to be better for highly accurate or traceable dimensional measurements. It can also be performed on the machine, and the method is convenient for observing variation in the tip geometry.

In the following experiments, both SEM imaging and a tip characterizer are used for tip geometry measurement. SEM images of the tips are taken before and after the roughness measurements. A tip characterizer developed by PTB is used to map the tip geometry variation during the repeated roughness measurements. Additionally, a tip geometry evaluation method suitable for blunt tips is proposed and applied.

### 2.1. Parameters to Define the Tip Geometry in 2D

A conical or pyramidal tip is often defined by two parameters: tip radius r and opening angle θ, as shown in Figure 4. The 2D tip form is the tip outer contour in the traverse direction. The tip end of the 2D tip form is assumed to be circular. The tip radius is the radius of the fitted circle. The opening angle, also called the cone angle, is the angle of the fitted cone.

For a tip with nonideal form, the two parameters can only depict the tip geometry approximately. With sufficient sampling points, a least squares fitting lowers the influence of the measurement noises and provides reasonable reliability.

### 2.2. Tip Characterizer with Rectangular Structures

As described above, a tip characterizer is an artifact with specially designed features. These features may include sharp edges or features of known size. The rectangular structure is one of the most commonly used features in tip characterization. It can be used to map the tip outline and characterize both the tip radius and the opening angle.

The rectangular features are composed of rectangular ridges and grooves. Since the procedures for characterizing the tip using the rectangular ridge feature and the rectangular groove feature are similar, the characterizing procedure is explained here using an example of a rectangular groove feature with width *W* and height *H*, as shown in Figure 5. The measured profile is the trace of the tip apex (the dashed line), and the tip at different positions is drawn with dotted lines (Figure 5a). After acquiring the measured profile of the rectangular groove feature, both ridge edges A and B are determined (Figure 5b). Then the 2D tip form is drawn by first selecting the segment between edges A and B from the measured profile (Figure 5c), switching the segment parts on either side of the line perpendicular to the lowest point, and finally overlapping the A and B edges to acquire the tip geometry (Figure 5d).

### 2.3. Methods of Determining the Ridge Edges

Determining the positions of ridge edges A and B is important in tip characterization with a rectangular feature. It yields the tip apex and the width of the tip, as the result decides the uncertainties of the tip radius.

The slope of the measured profile *gr*(*x*) (i.e., the gradient of the profile) is used as the criterion in determining the edge positions:(1)$$gr\left(x\right)=\frac{dz\left(x\right)}{dx}$$
where *x* is the abscissa and *z* the measured profile.

It is assumed that the ridge top is flat and the gradients of the measured profiles on the ridge top are constant.

If the gradients on the ridge top are within the range *gr_t_* ± Δ*gr*, ridge edge positions A and B can be determined by searching for the first point where the profile gradient is out of this range:(2)$$\{\begin{array}{c} \left|gr\left(x\right)-gr_{t}\right|\leq \Delta gr\;x\langle A\;or\;x\rangle B\\ \left|gr\left(x\right)-gr_{t}\right|>\Delta gr\;x=A\;or\;x=B \end{array}$$

This method, which we here call the gradient range method, requires no additional information and is the most commonly used searching algorithm. However, it can only be applied to sharp tips. Errors can occur with a blunt tip since there may be no obvious gradient change at the ridge edge positions, and the wrong edge positions will be found.

A ridge determination method suitable for a blunt tip, named structure width method by us, is proposed. This method demands the knowledge of the groove width *W*.

This structure width method is illustrated in Figure 6. After leveling on the ridge top, the absolute gradients on the ridge top are close to zero. They are smaller than on the falling edges. The profile segment covering all the falling edges, beginning at edge A and ending at edge B, has a larger sum of absolute gradients than do other profile segments with length *W*.

With the structure width method, first-order leveling is performed on the ridge tops, and the ridge tops become horizontal (Figure 6a). *GR*(*x*), the absolute gradient sum of a profile segment with length *W*, is then calculated (Figure 6c):(3)$$GR\left(x\right)=\sum_{x}^{x+W}\left|gr\left(x\right)\right|$$

The curve of *GR*(*x*) is drawn (Figure 6d), and the position with the maximal *GR* is edge A. The structure width method places no requirement on the tip sharpness.

Because the microprobe tip can become blunt and flat during roughness measurements, the structure width method was chosen for our tip geometry evaluation. The tip characterizer structure widths were first measured by an atomic force microscope (Cypher, Oxford Instruments, Oxford, UK) with about 10 nm measurement uncertainty.

### 2.4. Tip Characterizer TSPN

A comb-shaped tip characterizer developed by PTB and known as TSPN is used in the experiment. There are three row structures on the artifact designated A, B, and C, as shown in Figure 7a. Row A is composed of three parts, with 10 groups in each part and 10 grooves in each group. The grooves in each group are separated by 3 µm wide ridges, and all grooves in a group have the same width. The group groove width increases from 0.1 µm to 3 µm from left to right. The groups are separated by a 5 µm wide groove. The parts are separated by two 30 µm wide ridges and three 5 µm wide grooves.

Row B has alternating 3 µm wide ridges and 1 µm wide grooves.

Row C contains eight parts. In every part, the groove width changes from 0.1 µm to 3 µm, and grooves are separated by 3 µm wide ridges (see Figure 7b). Part separation is the same as in row A, as shown in Figure 7c.

The ridge height (groove depth) of all structures is 2 µm.

Grooves of different widths can characterize tips with various radii and heights.

In the experiments, the 5 µm wide groove in the middle of the group separation of row C was chosen to characterize the microprobe tip. Based on experience from our previous measurements, the width of a flat tip is expected to increase to several micrometers after 100 m of roughness measurement, as shown in Figure 2b. A wider groove is therefore more suitable for tip characterization, since if the tip end is wider than the groove, the tip will not be able to reach down to the groove’s falling edges, and no effective tip characterization can be performed.

## 3. Experiments and Analysis

The microprobe tip geometry variation experiments are carried out with Profilscanner, a metrological profiler developed in-house at PTB [24,25,26].

The experimental setup is shown in Figure 8. On the head of the Profilscanner is a 3D piezo motion stage (PI, model P-628.2CD for XY axes and P-622.ZCD for Z axis) with a travel range of 800 µm × 800 µm × 250 µm (X × Y × Z). The microprobe is glued and bonded on a holder and then mounted under the piezo stage with a 15° tilting angle. Three laser interferometers (SIOS, model SP2000) measure the movement of the microprobe tip and deliver traceable results.

On the base of the Profilscanner, a rotation stage for rotating the artifact around the Z axis is placed on a 3D coarse positioning stage with a motion range of 25 mm × 25 mm × 12 mm (X × Y × Z) (Newport stage M-436 and DC motor LTA-HS for XY axes and PI M-501.1PD for Z axis, not shown in Figure 8). A roughness standard is fixed on the rotation stage with its center coinciding with the center of the rotation stage. The tip characterizer TSPN is mounted next to the roughness standard. After sliding over the roughness standard, the microprobe is moved to the tip characterizer, and tip characterization is performed.

Steel roughness standards with arithmetical mean deviation Ra values from 0.02 µm to 0.1 µm are used in the experiments because roughness surfaces in this range are mainly considered in this work.

Before a new microprobe tip is put into use, it is imaged with an SEM (Hitachi, model TM4000Plus). Then it is mounted on the head of the Profilscanner, and the microprobe signal sensitivity is calibrated.

In signal sensitivity calibration, the piezo stage moves the microprobe downwards with the tip touching the structureless smooth surface on the tip characterizer. The sensitivity is acquired by calculating the slope between the microprobe’s output voltage and the vertical displacement of the piezo stage. With the signal sensitivity calibrated, the microprobe can be used in dimensional measurements. The maximum contact force during sensitivity calibration is approximately 50 µN.

After calibration is completed, the microprobe is moved to the tip characterizer TSPN, and the tip geometry is characterized. The tip traverses perpendicularly to the direction of the tip characterizer features. For an accurate tip geometry, the tip should keep unchanged in the tip characterization. Because both the sharp microprobe tip and the TSPN edges are fragile, the tip traverses the characterizer quite slowly at 1 µm/s and with a small probing force of 15 µN to avoid damage to the tip and the tip characterizer. The tip geometry obtained in this step is the tip geometry at 0 m measuring length shown in the figures below.

Comparing the tip geometry at 0 m (red line) and the SEM image in Figure 9, the tip radius at 0 m is larger than that in the SEM image. This means that the sensitivity calibration caused a loss of tip volume and increased the tip radius from 0.1 µm to 0.3 µm. The 50 µN contact force applied during sensitivity calibration takes about 0.5 µm off the tip end. The tip volume loss caused by sensitivity calibration occurred on all the microprobe tips used in the experiments.

After the above steps, the tip is moved such that it slides over the roughness standard surface with a constant probing force. The tip geometry is characterized every 0.25 m for the first 1 m of sliding and then every 0.5 m for the next 2 m of sliding. Thereafter, the tip characterization interval is determined by the observed tip geometry variations. The more stable the tip geometry, the longer the tip characterization interval.

Three experiments, one investigating the tip wear rate, one the influence of probing force, and one the tip-sample movement, are each performed using a new microprobe.

### 3.1. Tip Wear Rate

According to Archard’s wear law [27], the wear volume loss Δ*V_w_* is a first-order model of the wear rate *k_w_* and the normal load *F* over the sliding distance *d*:(4)$$\Delta V_{w}=k_{w}Fd$$

If the tip radius of an octagonal pyramidal tip increases from *r*_0_ to *r*_1_ because of tip volume loss, the tip volume loss Δ*V* is:(5)$$\Delta V=\frac{8\sin\left(22.5{}^{\circ}\right)\cos\left(22.5{}^{\circ}\right)\left(r_{1}^{3}-r_{0}^{3}\right)}{3tg\left(\frac{\theta }{2}\right)}$$

To investigate the tip wear rate, tip fracture should be avoided as much as possible. In the experiment, the rotation stage rotates the roughness standard such that the tip slides on the roughness surface with a constant linear speed of 420 µm/s. To ensure stable relative lateral movement of the tip and sample, the roughness standard begins rotating before the microprobe tip approaches the standard’s surface, and the probing force is kept constant at 50 µN. When the sliding motion is completed, the microprobe tip is lifted, while the standard continues to rotate.

A roughness standard with an arithmetical mean deviation Ra = 0.05 µm is used in this experiment.

Because of the debris produced during the experiment between the tip and the surface, the tip wear rate would be increased if the tip were to slide repeatedly over the same area. For this reason, the piezo scans the tip at a very slow speed of 0.1 µm/s and 400 µm scan range to make the tip slide over different areas of the roughness standard.

After 260 m of sliding, there was almost no change in the tip geometry. The tip radius remained at about 0.3 µm, as shown in Figure 9. The tip wear rate k_w_, calculated according to Equations (4) and (5), was approximately 5 × 10^−18^ m^2^N^−1^.

This result demonstrates that the tip volume loss caused by wear is quite small. According to the above calculated wear rate, the tip sliding distance would need to exceed 50 km at a probing force of 50 µN in order to increase the microprobe tip radius from 0.3 µm to 2 µm. This result does not align with our observations from roughness measurements. In normal roughness measurements, just several hundred meters of sliding is needed to increase the tip radius from 0.3 µm to 2 µm. This implies that most of the tip volume loss observed in roughness measurements is not due to wear but rather to fracture.

### 3.2. Influence of the Probing Force

In this experiment, the probing force is changed, and a roughness standard with Ra = 0.02 µm is used. Other conditions are kept the same as in the experiment investigating the tip wear rate. The roughness standard is changed so that the influence of a small difference of surface roughness on the tip wear rate can also be investigated.

Like the wear rate *k_w_*, the tip volume loss rate *k* is defined to be the tip volume loss Δ*V* caused by the normal load *F* over the sliding distance *d*:(6)$$k=\frac{\Delta V}{Fd}$$

After sensitivity calibration and 0 m tip geometry characterization, the tip slides over the roughness surface for 10 m with a probing force of 50 µN applied. As shown in Figure 10, the tip geometry remains relatively constant after 10 m of sliding.

In step 2, the tip approaches the surface with an increased probing force of 150 µN and is then lifted. The tip characterization result indicates that the tip radius increased from 0.3 µm to about 0.7 µm. The tip lost approximately 1 µm of height under the 150 µN probing force.

In step 3, the tip slides over the roughness surface for about 400 m with the probing force set to 50 µN. The tip geometry variation is quite small after 400 m of sliding, and the tip volume loss rate *k* is about 7 × 10^−18^ m^2^N^−1^. This tip volume loss rate is of the same order as the result obtained in the wear rate experiment discussed above. It demonstrates that a difference in the artifact’s surface roughness in the range of tens of nm has no great influence on the tip volume loss rate.

The probing force is then increased to 150 µN, and the tip slides another 300 m. The tip radius increases from 0.7 µm to about 1.2 µm. The volume loss rate is about 10^−16^ m^2^N^−1^, more than 10 times the volume loss rate in step 3. The increased tip volume loss rate indicates that the increased probing force causes not only more tip wear, but also tip fracture.

The tip volume loss as a function of the sliding distance is shown in Figure 11.

### 3.3. Influence of the Tip-Sample Movement

Unlike in the above experiments where tip sliding is achieved by rotating the roughness standard, the roughness standard here remains fixed, and the tip is moved by the piezo stage. The roughness standard with an arithmetical mean deviation of Ra 0.05 µm is used again here.

The SEM image implies that the tip used in this experiment was not manufactured perfectly, and there is a protrusion at the point about 1 µm away from the tip end, as shown in Figure 12. Because the tip under the protrusion cannot touch the tip characterizer, the characterized tip geometry at this position is wider than in the SEM image.

In step 1, the piezo stage scans the microprobe tip in one direction with a probing force of 50 µN, a scanning speed of 100 µm/s, and a scanning range of 750 µm. After the tip slides 750 µm on the roughness surface, it is lifted and moved to the start position of the next scan line without touching the surface. Then it approaches and slides on the surface again. The distance between the scan lines is 1 µm.

The characterized tip radius at 0 m is about 0.15 µm (the tip volume loss of this microprobe in the sensitivity calibration was less than in the previous experiments). The tip radius increases to about 0.3 µm after 0.5 m of one-way scanning and further to about 0.4 µm after another 1 m of scanning. The tip volume loss rate in this step is about 2 × 10^−15^ m^2^N^−1^.

In step 2, the piezo stage scans the microprobe tip sliding back and forth on the roughness surface. The probing force, scanning speed, scanning range, and line distance are kept the same as in step 1.

After half a meter of two-way scanning, the protrusion has disappeared from the characterized tip geometry. This implies that at least 1 µm of the tip end has broken off. The tip radius increases to 1.6 µm after another 3 m of scanning, which means that the tip has become about 4 µm shorter. The tip volume loss rate in this step is about 6 × 10^−14^ m^2^N^−1^_._

### 3.4. Analysis

The tip volume loss rates observed in the three experiments are shown in Figure 13. If the tip radius increases from 0.3 µm to 2 µm, it can slide more than 50 km while applying 50 µN of probing force during rotary roughness standard measurement. This sliding distance decreases to about 1.4 km when the probing force is increased to 150 µN. About 200 m of sliding is possible when one-way tip scanning is performed at 50 µN probing force, and only about 6.5 m of measurement can be performed if 50 µN of probing force is applied during back-and-forth tip scanning.

The conditions in tip wear rate experiments (WEs) and tip-sample movement experiments (MEs) are almost the same except as concerns the relative tip-sample movement and the tip sliding speed. The tip sliding speed in MEs is even lower than in WEs, but the tip volume loss rate in MEs is 300–9000 times more than that seen in WEs. The difference between the two types of experiments lies in the tip-sample relative lateral speed, which is constant in WEs during the entire sliding period, but not in MEs. In the scan start positions, the lateral relative speed increases from 0 to 100 µm/s within milliseconds, and vice versa in the scan end positions. The acceleration at these positions is tens of mm/s^2^. The lateral force on the tip end (i.e., the dynamic and static friction) changes dramatically, leading to tip fracture.

Si is hard but brittle in a room temperature air environment. The fracture toughness is quite low, about 1.28$\;\mathrm{MPa}\sqrt{m}$ in single-crystal silicon thin films [28]. By comparison, the fracture toughness of soda–lime glass is 0.7–0.8 $\mathrm{MPa}\sqrt{m}$.

In scanning mode, the suddenly changed lateral force has an impact on the tip and results in severe tip fracture. It indicates that the lateral dynamic load is damaging to the tip geometry stability.

The experiment results indicate that:(1)Tip wear in roughness measurements has only a minor influence on tip geometry variation. Through careful control of the probing force and the lateral force to reduce tip fracture, the microprobe tip can slide tens of kilometers with the tip radius remaining below 2 µm.(2)Tip fracture causes the most tip volume loss in roughness measurement. The lateral dynamic load on the tip, rather than the probing force, has a major influence on the tip fracture.

Determining the main reason of the tip geometry variation helps the researchers to seek the direction to develop a durable tip. Hard coating on tips is a common solution to protect the tip from abrasion. Because tip volume loss and geometry variation in roughness measurements are primarily due to tip fracture instead of tip wear, a nm thick hard coating cannot protect the microprobe tip from tip volume loss. For a geometrically stable tip, other solutions should be considered, such as changing the tip-sample relative movement to reduce the lateral dynamic load on the tip or using other tip materials with better fracture toughness.

## 4. Summary

Being a probe for high-speed roughness measurements, not only superior dynamic properties and qualified signal accuracy but also a geometrically stable tip is demanded to ensure reliable measurement results. A microprobe with an integrated silicon tip is a candidate for high-speed roughness measurements. However, the tip geometry is changed considerably during the measurements.

To investigate the main factors that contribute to microprobe tip volume loss in roughness measurements, a tip characterizer TSPN with rectangular structures was used to characterize tip geometry variation, and a method for reconstructing the tip geometry from the measured profiles was introduced.

Experiments were then conducted to explore the influences on tip geometry variation exerted by tip wear, probing force, and relative tip-sample movement. The results indicate that tip fracture and not tip wear causes most of the tip volume loss observed in roughness measurements. In scanning mode, the lateral dynamic load caused by tip acceleration results in severe tip fracture. Maintaining a constant tip-sample relative speed to avoid lateral dynamic load can serve to effectively protect the tip.

This investigation helps to seek the direction to develop a durable tip for high-speed roughness measurements.

## Figures and Tables

**Figure 1 sensors-22-01298-f001:**
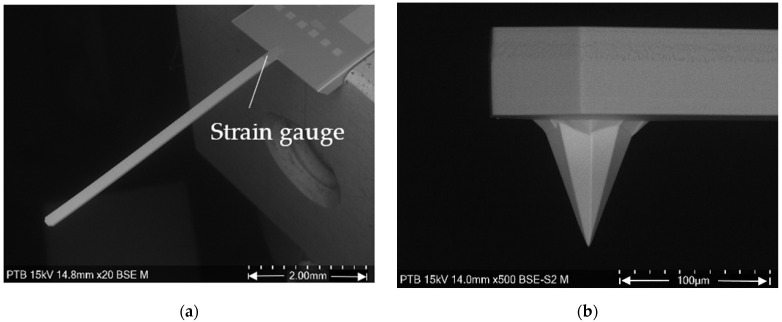
The 5 mm long piezoresistive microprobe. (**a**) Top view; (**b**) side view of tip.

**Figure 2 sensors-22-01298-f002:**
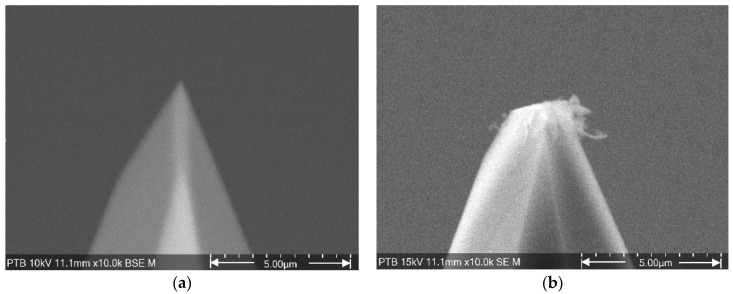
The microprobe tip geometry changes during roughness measurements. (**a**) New tip with a radius of approximately 0.1 µm; (**b**) the tip end becomes flat and wide after 300 m of roughness measurement.

**Figure 3 sensors-22-01298-f003:**
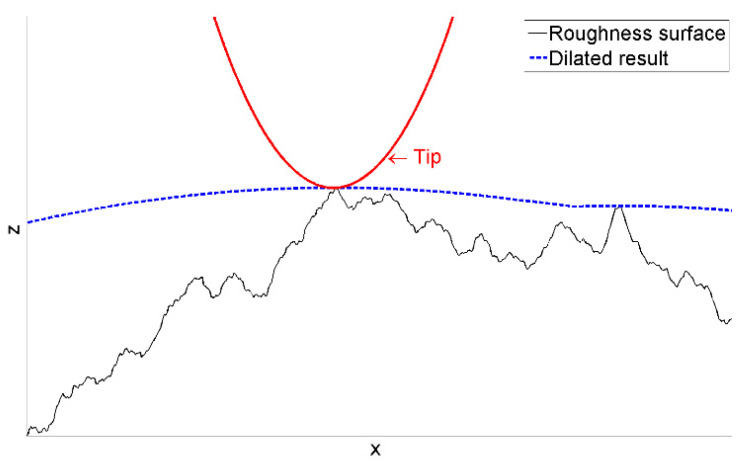
Schematic diagram of tip dilation on a rough surface.

**Figure 4 sensors-22-01298-f004:**
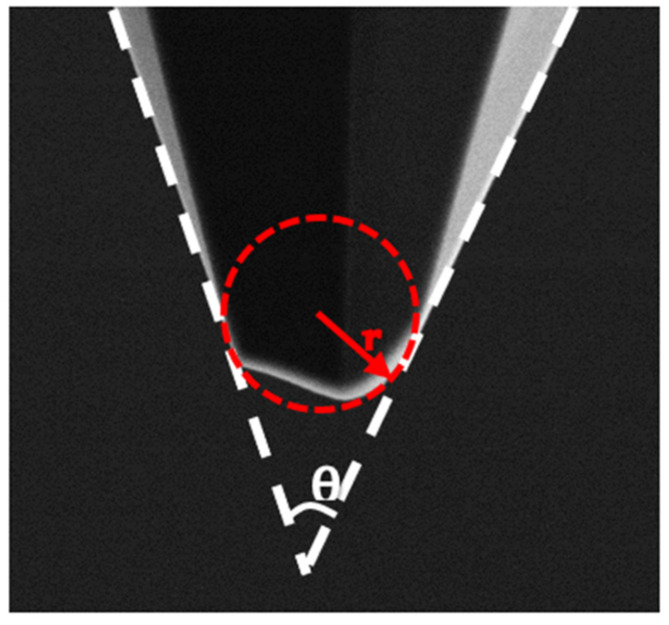
Tip radius r and opening angle θ define the geometry of the pyramidal tip.

**Figure 5 sensors-22-01298-f005:**
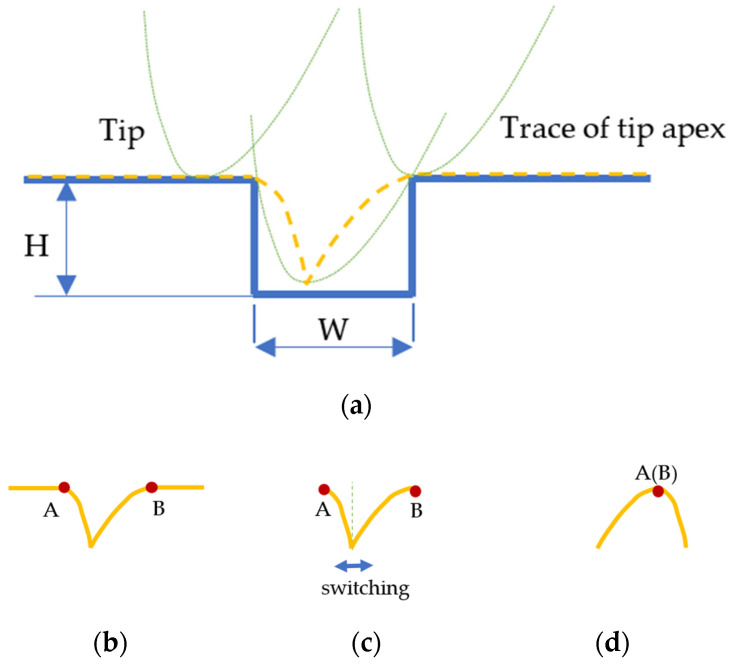
Tip characterization using a rectangular groove feature. (**a**) The measured profile of the rectangular groove feature; (**b**) determining ridge edges A and B; (**c**) selecting the segment between A and B from the measured profile and switching the segment parts on either side of the line perpendicular to the lowest point; (**d**) overlapping edges A and B and reconstructing the tip geometry.

**Figure 6 sensors-22-01298-f006:**
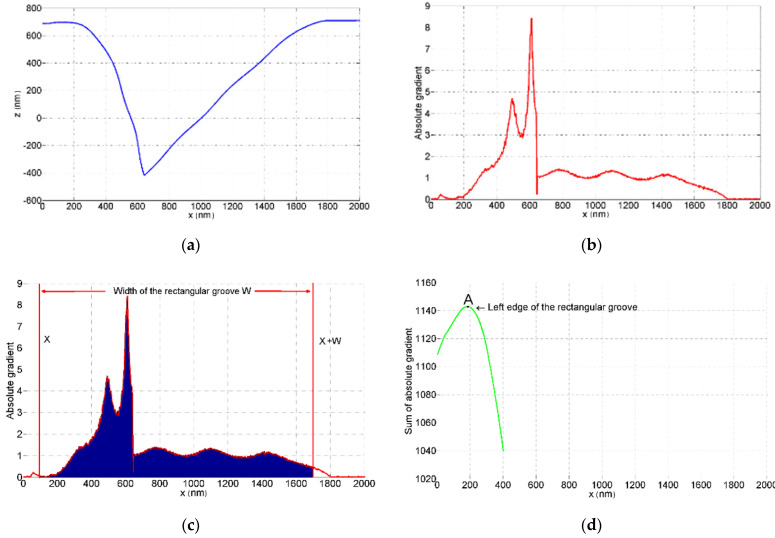
The structure width method is used to locate the ridge edges of the measured profile. (**a**) Measured profile of the rectangular groove structure after first-order leveling on ridge tops; (**b**) the absolute gradients of the profile |*gr*(*x*)|; (**c**) the sum of absolute gradients from *x* to *x* + *W* $GR\left(x\right)=\overset{x+W}{\underset{x}{\sum }}\left|gr\left(x\right)\right|$ (shaded area); (**d**) ridge edge A is the position with the maximal *GR* value; (**e**) ridge edge B is W away from ridge edge A.

**Figure 7 sensors-22-01298-f007:**
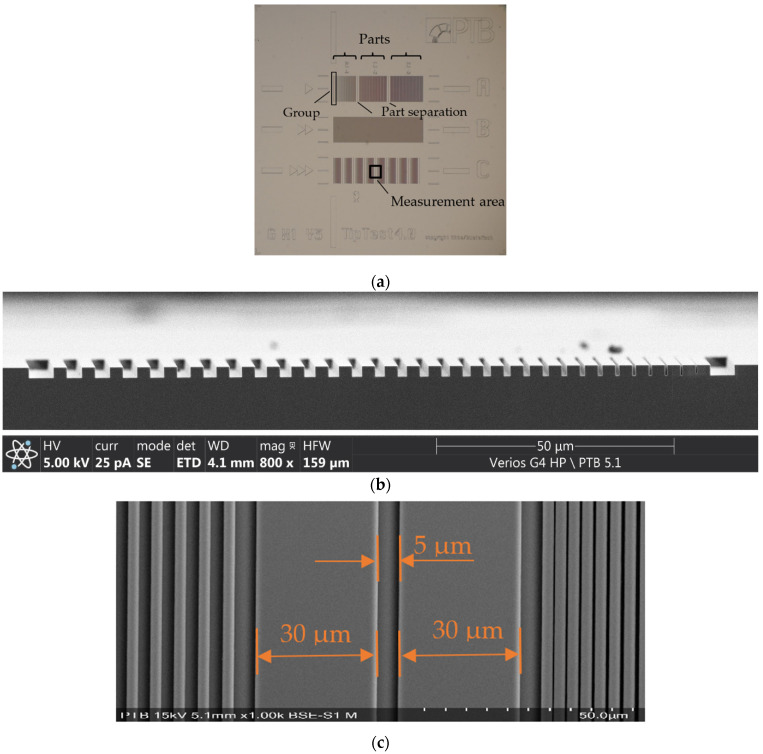
The TSPN tip characterizer used. (**a**) Three row structures, and the measurement area in the experiments; (**b**) cross section of row C, the groove widths from 0.1 µm to 3 µm; and (**c**) part separation structure of row C.

**Figure 8 sensors-22-01298-f008:**
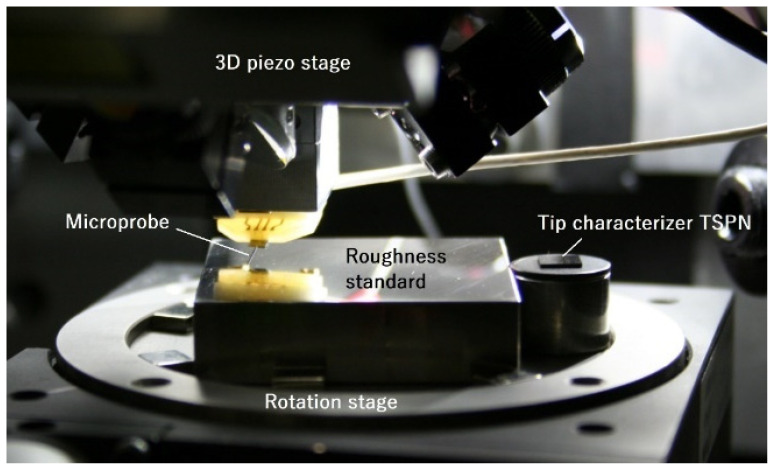
Experimental setup to investigate tip geometry variation during roughness measurements.

**Figure 9 sensors-22-01298-f009:**
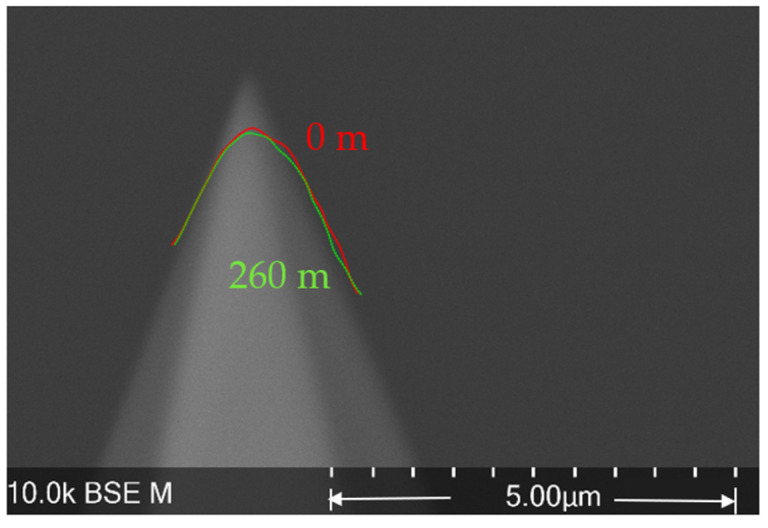
The tip geometry variation after 260 m of sliding over a roughness standard surface with Ra 0.05 µm.

**Figure 10 sensors-22-01298-f010:**
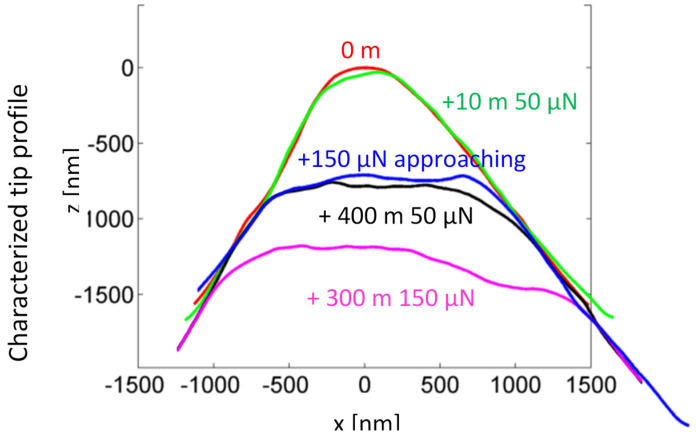
The tip profile changes with the probing force and the sliding distance.

**Figure 11 sensors-22-01298-f011:**
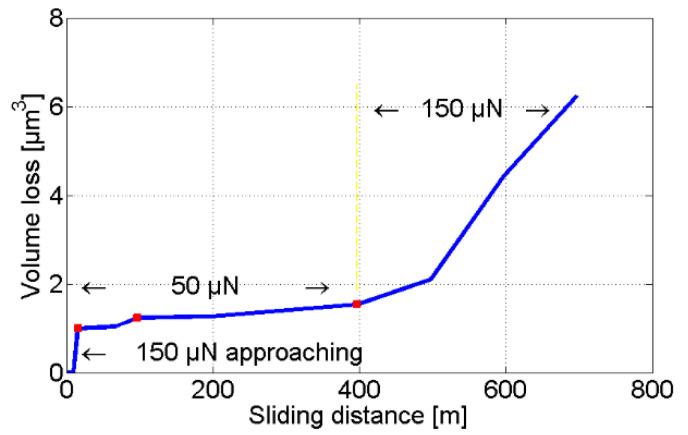
Tip volume loss plotted against sliding distance.

**Figure 12 sensors-22-01298-f012:**
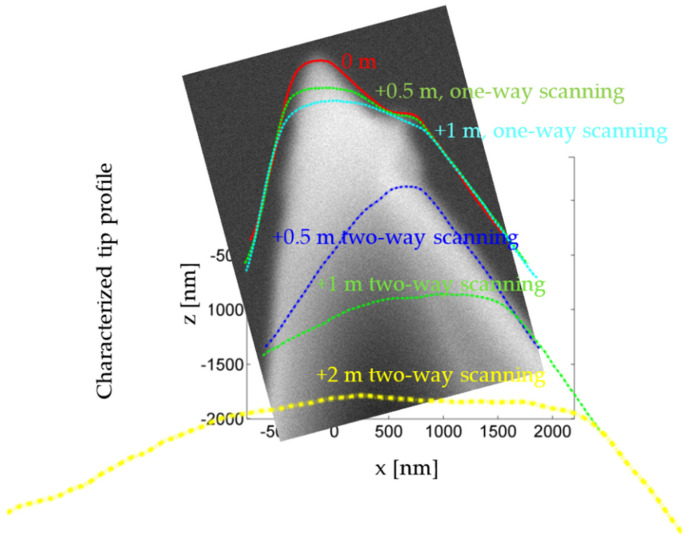
Change in tip profile during tip scanning.

**Figure 13 sensors-22-01298-f013:**
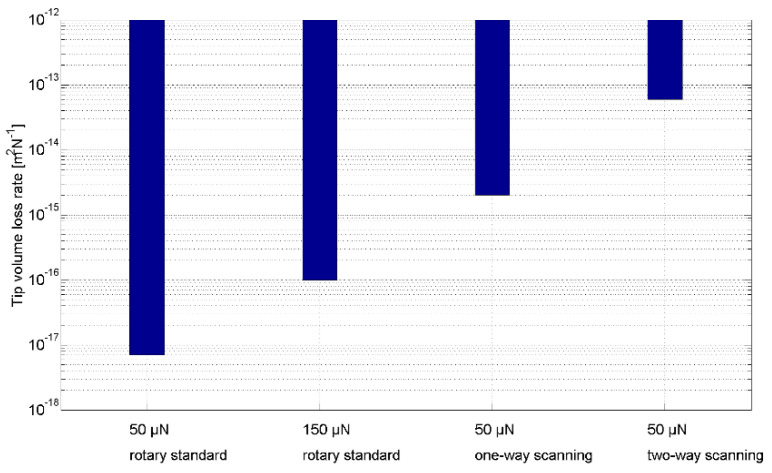
Tip volume loss rate under different experimental conditions.

## Data Availability

All data and code will be made available on request to the correspondent author’s email with appropriate justification.

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
