# Peer review of "Using a Tip Characterizer to Investigate Microprobe Silicon Tip Geometry Variation in Roughness Measurements"

_sensors, 2022, doi:10.3390/s22031298_

Round 1

Reviewer 1 Report

Dear authors, please find below some suggestions that must be considered for improvement of the manuscript submitted:

  1. Citing the sentence ‘It has been proven that the stylus method is more accurate than VSI for hard material roughness measurements [6].’ in fact, stylus methods seems to be more accurate than an optic, nevertheless, it should be indicated with more convincing references than mentioned [6]. Please propose more citations. Look below for some examples:

(1) Chin Y. Poon, Bharat Bhushan. Comparison of surface roughness measurements by stylus profiler, AFM and non-contact optical profiler. Wear, 1995, 190, 1, 76-88.

(2) Pawlus P. The errors of surface topography measurement using stylus instrument. Metrol. Meas. Syst. 2002, 3(9), 273–289.

(3) Podulka, P. Improved Procedures for Feature-Based Suppression of Surface Texture High-Frequency Measurement Errors in the Wear Analysis of Cylinder Liner Topographies. Metals 2021, 11, 143.

  1. There are too many self-citations, especially in line 53 (from [9] to [15]) when 2, maximum 3, the ‘most important’ should be selected and mentioned.
  2. There should be no advertising (lines 54-55) that the main thing is to present results with newly proposed methods that those already published. If the link to the website is allowed, it should be referenced, not located in the main (body) text of the reviewed manuscript.
  3. When high-speed surface topography measurement is applied, e.g. speed 10 mm/s (lines 61-62), what about a tip flight? There is no introduction to this point. The influence of stylus tip flight should be even mentioned. Please look for some previous examples:

(4) Song, J., Vorburger, T.V. (1996). Stylus flight in surface profiling. ASME J. Manuf. Sci. Engr., 118, 188–196.

(5) Pawlus P., Śmieszek M. The influence of stylus flight on change of surface topography parameters. Precision Engineering 2005, 29/3, 272-280.

  1. Method(s) that has been newly proposed in the manuscript (lines 94-95) should be highlighted that, simultaneously, novelty can be found difficult to be defined for a regular reader. It must be mentioned if a procedure is a novelty, if not, must be cited from the first source.
  2. According to the previous comment (no 5) and with the sentence from lines 108-109, if applied methods were compared with those previously cited (lines 96 and 97), it should be indicated.
  3. Directly from the sentence in lines 115-117, how were the fittings (circle and cone) performed? Are there any algorithms providing fitting with an appropriate accuracy? Was this accuracy sufficient? If a fitting accuracy is lost, results also can be not thoroughly convincing. Moreover, are there any disturbances that may cause distortions in the results from the fitting process? There are no words against measurement uncertainty or measurement noise. Please look for some examples:

(6) Podulka, P. Reduction of Influence of the High-Frequency Noise on the Results of Surface Topography Measurements. Materials 2021, 14, 333.

(7) Giusca C.L., Leach R.K., Helary F., Gutauskas T., Nimishakavi, L. Calibration of the scales of areal surface topography-measuring instruments: part 1. Measurement noise and residual flatness. Measurement Science and Technology, 2012, 23(3), 035008.

  1. If the ‘method of determining ridges edges’ (line 144) is not newly proposed by authors and, respectively, equations from (1) to (3) were described previously, original papers should be referenced.
  2. Fulfilling of the sentence ‘Based on experience from previous measurement…’, line 198, it should be mentioned (cited) in which paper of the authors, results of the previous studies and, respectively, ‘prior experiments’ were presented.
  3. Presenting the sentence ‘…the width of a flat tip is expected to increase to several micrometers after 199 100 meters of roughness measurement.’ what was the speed, or even medium speed, of the measurement? Does the measurement velocity influence the wear of the tip? If not, it should be mentioned. If yes, some references would also be appreciated. It must be precise.
  4. It should be justified why an amount of the roughness (Ra) was selected from 0.02 μm to 0.2 μm. If there is any limitation as far as the measurement of the selected (chosen) device is concerned.
  5. In section ‘3.4. Analysis’ there are not both a discussion or critical point of view of a proposed analysis and, consequently, of the results obtained. Considering studies of the methods applied and, simultaneously, novelty proposed, there is no weak(s) of the experiment presented. In all of the studies provided by the scientists, some weak can be constantly defined.
  6. According to the previous remark (no 12), section 3.4. should be reconstructed to make a reader more convinced about the results presented. Moreover, 3 conclusions, presented at the end of this section should be described more consciously, that neither this part (lines from 373 to 387) nor conclusions (section 4) are not convincing and, at least in the current form, make confused, what is the ‘domain’ novelty of the studies performed.
  7. The ‘Summary’ section seems to be too ‘general’. More details might have been mentioned as well. Further, dividing conclusions into separated, respectively, numbered gaps might give a more convincing and clear improvement of the novelty presented. Please try to emphasize the novelty, separate general conclusion(s) from other proposals or analyses.

Generally, the manuscript study area is interesting, nevertheless, the manuscript, at least in its current form, seems to be confusing and not suitable for taking into consideration to be published in the Sensors journal.

Concluding, improvement(s) must be provided.

Author Response

Response to the Reviewer
The authors appreciate the time and effort that the reviewers dedicated to our manuscript and are grateful for the comments on and valuable improvements to our paper. Below is a point-by-point response to the reviewers’ comments and concerns. The page numbers, line numbers and citation numbers refer to the revised manuscript.

  1. 1. Citing the sentence ‘It has been proven that the stylus method is more accurate than VSI for hard material roughness measurements [6].’ in fact, stylus methods seems to be more accurate than an optic, nevertheless, it should be indicated with more convincing references than mentioned [6]. Please propose more citations. Look below for some examples:
    (1) Chin Y. Poon, Bharat Bhushan. Comparison of surface roughness measurements by stylus profiler, AFM and non-contact optical profiler. Wear, 1995, 190, 1, 76-88.
    (2) Pawlus P. The errors of surface topography measurement using stylus instrument. Metrol. Meas. Syst. 2002, 3(9), 273–289.
    (3) Podulka, P. Improved Procedures for Feature-Based Suppression of Surface Texture High-Frequency Measurement Errors in the Wear Analysis of Cylinder Liner Topographies. Metals 2021, 11, 143.

Authors: Two citations are added.
7. Vorburger, T. V., Rhee, H. G., Renegar, T. B., Song, J. F., Zheng, A. Comparison of optical and stylus methods for measurement of surface texture. Int J Adv Manuf Technol. 2007, 33, 110–118
8. Poon, C.Y.; Bhushan, B. Comparison of surface roughness measurements by stylus profiler, AFM and non-contact optical profiler. Wear 1995, 190, 76-88

2. There are too many self-citations, especially in line 53 (from [9] to [15]) when 2, maximum 3, the ‘most important’ should be selected and mentioned.

Authors: Two citations [11, 12] remain and others are deleted.

3. There should be no advertising (lines 54-55) that the main thing is to present results with newly proposed methods that those already published. If the link to the website is allowed, it should be referenced, not located in the main (body) text of the reviewed manuscript.

Authors: The link is deleted.

4. When high-speed surface topography measurement is applied, e.g. speed 10 mm/s (lines 61-62), what about a tip flight? There is no introduction to this point. The influence of stylus tip flight should be even mentioned. Please look for some previous examples:
(4) Song, J., Vorburger, T.V. (1996). Stylus flight in surface profiling. ASME J. Manuf. Sci. Engr., 118, 188–196.
(5) Pawlus P., Śmieszek M. The influence of stylus flight on change of surface topography parameters. Precision Engineering 2005, 29/3, 272-280.

Authors: The sentences (lines 59-63) are amended.
“The microprobe demonstrates superior dynamics because of its low mass. Theoretical analysis and experimental results indicate that it can track surfaces with steep features up to traverse speeds of 10 mm/s with high fidelity [14]. When the probing force is larger than 28 μN, the
microprobe can measure the surface of 10 μm amplitude and 11 μm wavelength with the traverse speed up to 15 mm/s without tip flight [15].”

5. Method(s) that has been newly proposed in the manuscript (lines 94-95) should be highlighted that, simultaneously, novelty can be found difficult to be defined for a regular reader. It must be mentioned if a procedure is a novelty, if not, must be cited from the first source.

Authors: The citations [18, 19] are added in line 97.

18 Montelius, L.; Tegenfeldt, J. O. Direct observation of the tip shape in scanning probe microscopy. Appl. Phys. Lett, 1993, 62, 2628-2630.

19 Lantz, M. A.; O’Shea, S. J.; Welland, M. E. Characterization of tips for conducting atomic force microscopy in ultrahigh vacuum. Rev. Sci. Instrum., 1998, 69, 1757-1764.

6. According to the previous comment (no 5) and with the sentence from lines 108-109, if applied methods were compared with those previously cited (lines 96 and 97), it should be indicated.

Authors: Sentences are amended (line 112-114)
“A tip characterizer developed by PTB is used to map the tip geometry variation during the repeated roughness measurements. And a tip geometry evaluation method suitable for blunt tips is proposed and applied.”

7. Directly from the sentence in lines 115-117, how were the fittings (circle and cone) performed? Are there any algorithms providing fitting with an appropriate accuracy? Was this accuracy sufficient? If a fitting accuracy is lost, results also can be not thoroughly convincing. Moreover, are there any disturbances that may cause distortions in the results from the fitting process? There are no words against measurement uncertainty or measurement noise. Please look for some examples:
(6) Podulka, P. Reduction of Influence of the High-Frequency Noise on the Results of Surface Topography Measurements. Materials 2021, 14, 333.
(7) Giusca C.L., Leach R.K., Helary F., Gutauskas T., Nimishakavi, L. Calibration of the scales of areal surface topography-measuring instruments: part 1. Measurement noise and residual flatness. Measurement Science and Technology, 2012, 23(3), 035008.

Authors: Sentences are added to line 121-123
“For a tip with non-ideal form, the two parameters can only depict the tip geometry approximately. With sufficient sampling points, a least squares fitting lowers the influence of the measurement noises and provides reasonable reliability. “

8. If the ‘method of determining ridges edges’ (line 144) is not newly proposed by authors and, respectively, equations from (1) to (3) were described previously, original papers should be referenced.

Authors: The structure width method is proposed by authors. Sentences are added to line 168-169
“A ridge determination method suitable for a blunt tip, named structure width method by us, is proposed. This method demands the knowledge of the groove width W. “

9. Fulfilling of the sentence ‘Based on experience from previous measurement…’, line 198, it should be mentioned (cited) in which paper of the authors, results of the previous studies and, respectively, ‘prior experiments’ were presented.
10. Presenting the sentence ‘…the width of a flat tip is expected to increase to several micrometers after 100 meters of roughness measurement.’ what was the speed, or even medium speed, of the measurement? Does the measurement velocity influence the wear of the tip? If not, it should be mentioned. If yes, some references would also be appreciated. It must be precise.???

Authors: For both the comments 9 and 10
The sentences “Based on experience from previous measurements, the width of a flat tip is expected to increase to several micrometers after 100 meters of roughness measurement” is to explain why we select 5 μm wide groove in our experiments, not narrower grooves. The reason lies in:
(1) A groove cannot characterize a tip wider than the groove itself.
(2) We have done many measurements in our labor and observed that the roughness measurement caused the tip blunt and wide. But these measurements have not been published.
The sentences are amended (line 205-207)
“Based on experience from our previous measurements, the width of a flat tip is expected to increase to several micrometers after 100 meters of roughness measurement, as shown in Figure 2(b).”

11. It should be justified why an amount of the roughness (Ra) was selected from 0.02 μm to 0.2 μm. If there is any limitation as far as the measurement of the selected (chosen) device is concerned.

Author: Sentences in line 238-240 are amended
“Steel roughness standards with arithmetical mean deviation Ra values from 0.02 μm to 0.1 μm are used in the experiments because roughness surfaces in this range are mainly considered in this work.”

12. In section ‘3.4. Analysis’ there are not both a discussion or critical point of view of a proposed analysis and, consequently, of the results obtained. Considering studies of the methods applied and, simultaneously, novelty proposed, there is no weak(s) of the experiment presented. In all of the studies provided by the scientists, some weak can be constantly defined.

Author: For the weak of the experiment, the sentences are amended (line 252-255)
“For an accurate tip geometry, the tip should keep unchanged in the tip characterization. Because both the sharp microprobe tip and the TSPN edges are fragile, the tip traverses the characterizer quite slowly at 1 μm/s and with a small probing force of 15 μN to avoid the damage to the tip and the tip characterizer.”

13. According to the previous remark (no 12), section 3.4. should be reconstructed to make a reader more convinced about the results presented. Moreover, 3 conclusions, presented at the end of this section should be described more consciously, that neither this part (lines from 373 to 387) nor conclusions (section 4) are not convincing and, at least in the current form, make confused, what is the ‘domain’ novelty of the studies performed.

Author: Sentences are amended (line 383-403)
“In scanning mode, the suddenly changed lateral force impacts on the tip and results in severe tip fracture. It indicates that the lateral dynamic load is damaging to the tip geometry stability.
The experiment results indicate that:
(1) Tip wear in roughness measurements has only a minor influence on tip geometry variation. Through careful control of the probing force and the lateral force to reduce tip fracture, the microprobe tip can slide tens of kilometers with the tip radius remaining below 2 μm.
(2) Tip fracture causes the most tip volume loss in roughness measurement. The lateral dynamic load on the tip, rather than the probing force, has a major influence on the tip fracture.
Determining the main reason of the tip geometry variation helps the researchers to seek the direction to develop a durable tip. Hard coating on tips is a common solution to protect the tip from abrasion. Because tip volume loss and geometry variation in roughness measurements
are primarily due to tip fracture instead of tip wear, a nm thick hard coating cannot protect the microprobe tip from tip volume loss. For a geometrically stable tip, other solutions should be considered, such as changing the tip-sample relative movement to reduce lateral dynamic load on the tip or using other tip materials with better fracture toughness.”

14. The ‘Summary’ section seems to be too ‘general’. More details might have been mentioned as well. Further, dividing conclusions into separated, respectively, numbered gaps might give a more convincing and clear improvement of the novelty presented. Please try to emphasize the novelty, separate general conclusion(s) from other proposals or analyses.

Author: The summary is amended (line 408-424)
“Being a probe for high-speed roughness measurements, not only superior dynamic properties and qualified signal accuracy, but also a geometrically stable tip is demanded to ensure reliable measurement results. Microprobe with an integrated silicon tip is a candidate for high-speed roughness measurements. However, the tip geometry is changed considerably during the measurements.
To investigate the main factors that contribute to microprobe tip volume loss in roughness measurements, a tip characterizer TSPN with rectangular structures was used to characterize tip geometry variation and a method for reconstructing the tip geometry from the measured profiles was introduced.
Experiments were then conducted to explore the influences on tip geometry variation exerted by tip wear, probing force and relative tip-sample movement. The results indicate that tip fracture and not tip wear causes most of the tip volume loss observed in roughness measurements. In scanning mode, the lateral dynamic load caused by tip acceleration results in severe tip fracture. Maintaining a constant tip-sample relative speed to avoid lateral dynamic load can serve to effectively protect the tip.
This investigation helps to seek the direction to develop a durable tip for high-speed roughness measurements. “

Reviewer 2 Report

The authors present important issue concerning investigation of microprobe’s geometry variation. The paper is of a high level both in terms of technical content and form. There are few minor issues that should be taken into consideration in the final version.

Line 55 : Please move the webpage of CIS to the References and cite it in line 55. Then you will avoid large gaps in line 54. By the way - please check this link, probably something is wrong with it since some error on the page occurs.

Fig. 1, 2, 9: I have general remark to figures taken from microscope. Some readers may be confused by the marking of the scale in these figures. The value shown in the lower right corner corresponds to not one but ten divisions. I propose to add suitable information in the caption of the figures or add an unambiguous marking of the scale (e.g as used in orange in Fig. 7c for 5 um)

Line 295: It is not clear why you changed the roughness standard in the experiment with probing source (tip wear 0.05 um, here 0.02 um)

Line 310 and Fig.10: You wrote that the geometry variation after 400 m is similar to the results obtained for the wear test, but in Fig. 9 geometry changes are not visible, in contrast to Fig.10. Could you please comment this.  

Line 369 and 370: Unit “MPa” and “m” should not be written in italic. Please correct it.

Author Response

Response to the Reviewer
The authors appreciate the time and effort that the reviewers dedicated to our manuscript and are grateful for the comments on and valuable improvements to our paper. Below is a point-by-point response to the reviewers’ comments and concerns. The page and line numbers refer to the revised manuscript.

Line 55 : Please move the webpage of CIS to the References and cite it in line 55. Then you will avoid large gaps in line 54. By the way - please check this link, probably something is wrong with it since some error on the page occurs.

Authors: The webpage of CIS is removed.
During the manuscript preparation, CIS deleted the link from its webpage. Sorry for the mistake that we didn’t check the link carefully. The authors thank the reviewer gratefully for the correction.

Fig. 1, 2, 9: I have general remark to figures taken from microscope. Some readers may be confused by the marking of the scale in these figures. The value shown in the lower right corner corresponds to not one but ten divisions. I propose to add suitable information in the caption of the figures or add an unambiguous marking of the scale (e.g as used in orange in Fig. 7c for 5 um)

Authors: The scales in Fig. 1, 2 and 9 are marked.

Line 295: It is not clear why you changed the roughness standard in the experiment with probing source (tip wear 0.05 um, here 0.02 um)

Authors: The explanation is added to Line 307-308:
“The roughness standard is changed so that the influence of a small difference of surface roughness on the tip wear rate can also be investigated.”

Line 310 and Fig.10: You wrote that the geometry variation after 400 m is similar to the results obtained for the wear test, but in Fig. 9 geometry changes are not visible, in contrast to Fig.10. Could you please comment this.

Authors: The tip wear rate is very small. The tip sliding distance in Fig. 9 is much shorter than that in Fig. 10, as the result the tip variation in Fig.9 is smaller and not visible.
The sentences are amended (line 321-322). “similar to” is changed to “the same order as”.
“The tip geometry variation is quite small after 400 m of sliding and the tip volume loss rate k is about 7 × 10-18 m2N-1. This tip volume loss rate is of same order as the result obtained in the wear rate experiment discussed above.”

Line 369 and 370: Unit “MPa” and “m” should not be written in italic. Please correct it.
Authors: They are corrected.

Round 2

Reviewer 1 Report

Dear authors, all of the raised issues were improved in a required manner. Therefore, manusript in the current, revised form, can be considered for publication in the Sensors journal.